# Inducible and Conditional Activation of Adult Neurogenesis Rescues Cadmium-Induced Hippocampus-Dependent Memory Deficits in ApoE4-KI Mice

**DOI:** 10.3390/ijms24119118

**Published:** 2023-05-23

**Authors:** Megumi T. Matsushita, Hao Wang, Glen M. Abel, Zhengui Xia

**Affiliations:** Department of Environmental and Occupational Health Sciences, School of Public Health, University of Washington, Seattle, WA 98105, USA

**Keywords:** adult neurogenesis, cadmium, neurotoxicity, Alzheimer’s disease, gene–environment interaction

## Abstract

The apolipoprotein E (ApoE) gene is a genetic risk factor for late-onset Alzheimer’s disease, in which ε4 allele carriers have increased risk compared to the common ε3 carriers. Cadmium (Cd) is a toxic heavy metal and a potential neurotoxicant. We previously reported a gene–environment interaction (GxE) effect between ApoE4 and Cd that accelerates or increases the severity of the cognitive decline in ApoE4-knockin (ApoE4-KI) mice exposed to 0.6 mg/L CdCl_2_ through drinking water compared to control ApoE3-KI mice. However, the mechanisms underlying this GxE effect are not yet defined. Because Cd impairs adult neurogenesis, we investigated whether genetic and conditional stimulation of adult neurogenesis can functionally rescue Cd-induced cognitive impairment in ApoE4-KI mice. We crossed either ApoE4-KI or ApoE3-KI to an inducible Cre mouse strain, Nestin-CreER^TM^:caMEK5-eGFP^loxP/loxP^ (designated as caMEK5), to generate ApoE4-KI:caMEK5 and ApoE3-KI:caMEK5. Tamoxifen administration in these mice genetically and conditionally induces the expression of caMEK5 in adult neural stem/progenitor cells, enabling the stimulation of adult neurogenesis in the brain. Male ApoE4-KI:caMEK5 and ApoE3-KI:caMEK5 mice were exposed to 0.6 mg/L CdCl_2_ throughout the experiment, and tamoxifen was administered once Cd-induced impairment in spatial working memory was consistently observed. Cd exposure impaired spatial working memory earlier in ApoE4-KI:caMEK5 than in ApoE3-KI:caMEK5 mice. In both strains, these deficits were rescued after tamoxifen treatment. Consistent with these behavioral findings, tamoxifen treatment enhanced adult neurogenesis by increasing the morphological complexity of adult-born immature neurons. These results provide evidence for a direct link between impaired spatial memory and adult neurogenesis in this GxE model.

## 1. Introduction

Alzheimer’s disease (AD) is a leading neurodegenerative disease characterized by progressive cognitive decline and memory loss. It is the leading cause of dementia, accounting for an estimated 60 to 80% of cases [1]. While amyloid precursor protein (*APP*), presenilin1 (*PSEN1*), and presenilin2 (*PSEN2*) genes have been identified as causal genes for early-onset AD, they account for less than 6% of AD cases [2,3]. A combination of environmental and genetic factors likely contributes to the disease etiology for sporadic, late-onset AD [4,5]. Furthermore, an interaction between genetic and environmental factors (gene–environment interaction, GxE) may lead to increased severity or acceleration of cognitive decline.

The apolipoprotein E (*APOE*) gene is the strongest known genetic risk factor for late-onset AD. Carriers of the ε4 allele have estimates of up to 12 times higher risk than carriers of the more common ε3 allele [2,6]. Cadmium (Cd) is a toxic heavy metal of public health concern due to ubiquitous exposure to the general population through food and smoking [7]. Epidemiology studies have reported associations between Cd and cognitive impairment in US adults [8,9,10]. We have demonstrated a causal relationship between Cd exposure and cognitive decline in a controlled mouse study [11].

Furthermore, our lab utilized a humanized knock-in (KI) mouse model of AD to investigate the effect of GxE on cognition. These mice express the human ε4 (ApoE4-KI) or human ε3 alleles (ApoE3-KI) under the control of the endogenous mouse ApoE promoter; thus, the human ApoE3 or ApoE4 are expressed at physiological levels [12]. Using this transgenic line, we reported that ApoE4-KI males had earlier onset of cognitive deficits compared to ApoE3-KI males with the treatment of 0.6 mg/L CdCl_2_ in drinking water [13]. While this study provided experimental evidence of a GxE effect between ApoE4 and Cd exposure, the underlying mechanisms remain undefined.

The process of adult neurogenesis involves the generation of neurons from a subset of neural progenitor cells (NPCs) in a quiescent state into fully differentiated neurons that integrate into existing neuronal circuits [14,15]. In the adult mammalian brain, the adult NPC (aNPC) population is found in two neurogenic regions: the subventricular zone, which gives rise to new inhibitory neurons in the olfactory bulb, and the subgranular zone of the hippocampal dentate gyrus (DG), which gives rise to excitatory granular cells in the DG [14,16]. Integrating adult-born neurons in the hippocampus is critical in learning and memory, with ablation or reduction of adult hippocampal neurogenesis leading to impairments in hippocampus-dependent learning and memory tasks [17,18,19]. We previously reported that Cd impairs adult hippocampal neurogenesis at exposure levels that induce cognitive deficits in mice [20,21].

The extracellular signal-regulated kinase 5 (ERK5), a member of the mitogen-activated protein kinase (MAPK) family, is specifically expressed in adult neurogenic regions and is critical for the regulation of adult hippocampal neurogenesis [22]. ERK5 is activated by MEK5, a specific upstream activating kinase for ERK5 but does not activate other members of the MAP kinases, including ERK1/2, JNK, or p38, even when overexpressed [23,24]. We generated Nestin-CreER^TM^:caMEK5-eGFP^loxP/loxP^ double transgenic mice (simplified to caMEK5 mice) in which a constitutively active MEK5 (caMEK5) is fused with an enhanced green fluorescent protein (eGFP). The expression of caMEK5-eGFP is controlled by the Nestin-CreER^TM^ promoter, induced upon tamoxifen administration, and limited to nestin-expressing neurogenic regions of the brain. This inducible expression of caMEK5 is sufficient to genetically and conditionally induce adult neurogenesis in the hippocampus [25]. Thus, this gain-of-function caMEK5 transgenic mouse line enables the investigation of a causal relationship between Cd impairment of adult neurogenesis and memory loss.

The main goal of this study is to investigate the molecular and cellular mechanisms underlying the GxE effect of ApoE4 and Cd on hippocampus-dependent memory. Our lab has previously demonstrated that Cd impairs adult neurogenesis [20] and established a causal link between Cd-induced impairments in hippocampus-dependent memory and adult neurogenesis [21]. We hypothesized that the GxE effect of ApoE4 and Cd on adult hippocampal neurogenesis contributes to the impairment of hippocampus-dependent learning and memory. To this end, we designed a functional rescue experiment utilizing a triple-transgenic mouse line to conditionally and genetically stimulate adult neurogenesis following the observation of hippocampus-dependent memory impairment in a GxE model of humanized ApoE4-KI:caMEK5 mice treated with Cd at environmentally relevant exposure levels.

## 2. Results

### 2.1. Mouse Body Weight and Water Consumption

We exposed 8- to 10-week-old male ApoE3-KI:caMEK5 and ApoE4-KI:caMEK5 animals to 0.6 mg/L CdCl_2_ through the end of the experiment (see Section 4). Tamoxifen and vehicle treatment groups were selected a priori. We used 0.6 mg/L CdCl_2,_ as previously described, to study the cognitive effects of Cd in mice at levels relevant to the general population [13,21].

We recorded the body weights of the mice every 1–2 weeks throughout the experiment to monitor the effects of tamoxifen on the body weights of mice from each genotype. We did not observe an effect of tamoxifen on body weight, (Figure 1A, mixed-effects linear regression: ApoE3-KI:caMEK5, F_(1, 26)_ = 1.14, *p* = 0.30; ApoE4-KI:caMEK5, F_(1, 25)_ = 0.0002, *p* = 0.99), and observed a significant interaction effect of experiment week and tamoxifen treatment (mixed-effects linear regression: ApoE3-KI:caMEK5, F_(52, 1309)_ = 9.61, *p* < 2 × 10^−16^ ApoE4-KI:caMEK5, F_(47, 1175)_ = 9.43, *p* < –2 × 10^−16^) with body weights of the tamoxifen-treated animals being significantly lower than the vehicle-treated animals in both genotypes temporarily (Welch’s two sample *t*-test, Tukey HSD corrected: ApoE3-KI:caMEK5: *p* < 0.05 for weeks 35–49; ApoE4-KI:caMEK5: *p* < 0.05 for weeks 28–30). This suggests that tamoxifen treatment has some reversible toxicity in mice, consistent with the literature [26,27]. We also recorded water consumption every week throughout the experiment and did not observe any effect of tamoxifen on water consumption (Figure 1B, mixed-effects linear regression: ApoE3-KI:caMEK5, tamoxifen: F_(1, 4)_ = 0.06, *p* = 0.82, experiment week x tamoxifen: F_(56, 224)_ = 0.86, *p* = 0.75; ApoE4-KI:caMEK5, tamoxifen: F_(1, 4)_ = 1.54, *p* = 0.28, experiment week x tamoxifen: F_(51, 202)_ = 1.25, *p* = 0.14). These data suggest that tamoxifen treatment has no effect on water consumption.

### 2.2. NOL Test

We performed a 1 h NOL test to assess hippocampus-dependent spatial memory (Figure 2A). Before Cd exposure, all groups of mice spent significantly more time exploring the object in the new location (location C) compared to the old location (location A) in the testing session, suggesting that all mice remembered the original object locations and thus were able to distinguish between the old and new locations (Figure 2B, Baseline). ApoE4-KI:caMEK5 mice started to show a spatial memory deficit at 16 weeks of Cd exposure, indicated by a failure to discriminate between old and new locations, and continued to show deficits at weeks 17, 18, and 20, whereas ApoE3-KI:caMEK5 mice started to show spatial memory deficits later, at weeks 28, 29, and 30 (Figure 2B, red arrowheads). This recapitulates the GxE interaction effect on the NOL memory reported in our previous study of ApoE3-KI and ApoE4-KI mice [13], suggesting an effect of ApoE4 genotype on Cd-induced hippocampus-dependent spatial memory even with the caMEK5 genetic background.

For each genotype, after confirming a Cd-induced memory deficit in the NOL test in at least three time points, we administered tamoxifen to the animals via oral gavage to conditionally and selectively induce caMEK5 expression in adult neural progenitor cells. Tamoxifen or vehicle treatment started at week 22 or 32 for ApoE4-KI:caMEK5 or ApoE3-KI:caMEK5, respectively. Animals were allowed to recover for 3.5–4 weeks after the last tamoxifen treatment. Tamoxifen-treated ApoE4-KI:caMEK5 mice exhibited a rescue effect in the NOL test behavior at experimental weeks 32.5, 34, 38.5, and 39.5 (i.e., 3.5, 5, 9.5, and 10.5 weeks after last tamoxifen treatment, respectively) while vehicle-treated ApoE4-KI:caMEK5 mice continued to exhibit a deficit in spatial memory (Figure 2B, top, purple arrowheads). This suggests that the inducible and conditional expression of caMEK5 reversed memory impairment in ApoE4-KI:caMEK5 mice, even in the continued presence of Cd exposure. Similarly, tamoxifen-treated ApoE3-KI:caMEK5 mice were able to discriminate between the old and new locations at experimental weeks 42, 45, and 47 (i.e., 4, 7, and 9 weeks after the last tamoxifen treatment, respectively), while vehicle-treated ApoE3-KI:caMEK5 mice continued to exhibit a deficit in spatial memory (Figure 2B, bottom, purple arrowheads). This suggests that the inducible and conditional expression of caMEK5 also rescued ApoE3-KI:caMEK5 mice from the Cd-induced impairment of hippocampus-dependent spatial working memory.

### 2.3. Blood and Cortex Cd Concentrations

We collected the blood and brain tissue for Cd analysis from the behavioral cohort at euthanasia to assess Cd exposure levels in ApoE3-KI:caMEK5 and ApoE4-KI:caMEK5 mice at experimental weeks 54 and 49, respectively. All samples had Cd levels above the detection limit. The end-of-experiment blood Cd levels were largely within 0.2–0.4 µg/L range, comparable to blood Cd levels found in the general US population (men: 0.206–0.255 µg/L; women: 0.263–0.304 µg/L, 2011–2018 geometric mean range [28]), with means ranging from 0.315–0.347 µg/L across the four treatment groups. There were no significant differences between vehicle and tamoxifen-treated groups in blood Cd of ApoE3-KI:caMEK5, blood Cd of ApoE4-KI:caMEK5, or brain Cd of ApoE3-KI:caMEK5 (Figure 3A–C). The brain Cd levels of tamoxifen-treated ApoE4-KI:caMEK5 mice were slightly higher compared to vehicle-treated ApoE4-KI:caMEK5 mice (Figure 3D, Welch’s two sample *t*-test: *p* = 0.03), with mean end of experiment brain Cd levels ranging from 1.68–2.01 pg/mg. These data suggest that tamoxifen treatment did not lower Cd levels in blood or brain, and that a reduction in Cd levels is not a likely reason for the observed behavioral rescue effect.

### 2.4. Locomotor Activity and Anxiety

We performed the open field test at baseline before Cd treatment to exclude any intrinsic differences in locomotor activity and anxiety between tamoxifen and vehicle groups within each genotype. There were no significant differences in the open-field locomotor activity between tamoxifen and vehicle groups in either genotype prior to the Cd exposure (Figure 4, baseline). After the NOL test confirmed the behavioral rescue of memory deficits, we also conducted the open field test to assess the effects of tamoxifen on locomotor activity and anxiety at experimental weeks 48 and 41 for ApoE3-KI:caMEK5 and ApoE4-KI:caMEK5, respectively. Tamoxifen-treated ApoE3-KI:caMEK5 mice spent more time moving and moved a longer distance compared to vehicle-treated ApoE3-KI:caMEK5 mice, while there were no significant differences between vehicle- and tamoxifen-treated ApoE4-KI:caMEK5 mice (Figure 4, after Cd + Tam or Veh). This suggests that tamoxifen treatment increased locomotor activity in ApoE3-KI:caMEK5 animals. While comparisons between ApoE3-KI:caMEK5 and ApoE4-KI:caMEK5 animals are inappropriate due to time point differences, tamoxifen-treated ApoE3-KI:caMEK5 animals’ locomotor activity were closer to the vehicle- and tamoxifen-treated ApoE4-KI:caMEK5 animals.

We used time and distance in the margin, center, or entries to the center to assess anxiety. There were no significant differences in anxiety behavior between tamoxifen and vehicle groups in either genotype prior to Cd exposure (Figure 5, baseline). There were also no significant differences between vehicle- and tamoxifen-treated ApoE4-KI:caMEK5 mice (Figure 5, after Cd + Tam or Veh). Together, the data in Figure 4 and Figure 5 suggest that tamoxifen had no effect on locomotor activity or level of anxiety in ApoE4-KI:caMEK5 mice.

At week 48, tamoxifen-treated ApoE3-KI:caMEK5 mice entered the arena center more frequently and moved longer distances in the arena margin compared to vehicle-treated ApoE3-KI:caMEK5 mice. Taken together with the observed increase in locomotor activity, these data suggest that tamoxifen decreased anxiety levels and increased locomotor activity in ApoE3-KI:caMEK5 mice.

### 2.5. caMEK5 Expression Stimulates Adult Hippocampal Neurogenesis in ApoE4 Mice

We investigated the effects of tamoxifen-induced expression of caMEK5 on adult hippocampal neurogenesis using a separate cohort of mice from the same breeding group as the behavior cohort. caMEK5 is sequence-tagged with eGFP in the transgenic mouse strain (see Section 4); therefore, we performed eGFP immunostaining of brain tissues collected from the cellular cohort to confirm caMEK5 expression. eGFP staining was found in the DG of tamoxifen-treated groups in each genotype but not in vehicle-treated controls (Figure 6). These data confirm that caMEK5 expression is conditionally induced by tamoxifen treatment in the DG of both ApoE3-KI:caMEK5 and ApoE4-KI:caMEK5 mice.

To investigate the effects of caMEK5 expression on aNPC survival in the DG, BrdU was administered to cellular cohort animals 2.5 weeks prior to tissue collection, which coincided with the confirmation of NOL rescue in tamoxifen-treated animals in the behavioral cohort (ApoE3-KI:caMEK5: experiment week 47; ApoE4-KI:caMEK5: experiment week 40). We first quantified the BrdU-retaining cells (surviving adult-born cells) and BrdU^+^ and NeuN^+^ cells (adult-born mature neurons). We observed no differences between tamoxifen and control groups of either genotype in the number of surviving adult-born cells, adult-born mature neurons, or the fraction of adult-born mature neurons over adult-born cells (Figure 7). We also quantified BrdU^+^ and doublecortin^+^ (DCX^+^) cells (adult-born immature neurons). We observed no differences between the tamoxifen and control groups of either genotype in the number of surviving adult-born cells, adult-born immature neurons, or the fraction of adult-born immature neurons over adult-born cells (Figure 8).

We further assessed the dendritic morphology, a measure of neuronal maturation, of adult-born immature neurons in the DG of the cellular cohort mice. To avoid damage to the neuronal processes during the acid treatment in BrdU staining, we performed separate DCX immunostaining without BrdU co-staining to visualize and trace neuronal processes of newly generated, immature neurons (Figure 9A and Figure 10A). Tamoxifen treatment significantly increased the total dendritic length in both ApoE3-KI:caMEK5 (Figure 9B; Welch’s two sample *t*-test, *p* = 0.00026) and ApoE4-KI:caMEK5 mice (Figure 10B: Welch’s two sample *t*-test, ApoE4-KI:caMEK5: *p* = 4.6 × 10^−6^). Dendritic crossings of tamoxifen-treated animals were significantly higher than vehicle-treated animals in both genotypes (Figure 9C and Figure 10C; ApoE3-KI:caMEK5: 20–60 µm from the soma; ApoE4-KI:caMEK5: 20–120 µm from the soma). For ApoE3-KI:caMEK5 mice, we did not observe an effect of tamoxifen on the number of dendritic crossings (mixed-effects linear regression: F_(1, 4.42)_, *p* = 0.70) but observed a significant interaction effect of radius and tamoxifen treatment (mixed-effects linear regression: F_(18, 1458)_ = 2.47, *p* = 0.00056). For ApoE4-KI:caMEK5 mice, tamoxifen increased the number of dendritic crossings (mixed-effects linear regression: F_(1, 95)_ = 22.3, *p* = 7.98 × 10^−6^), and there is a significant interaction effect of radius and tamoxifen treatment (mixed-effects linear regression: F_(18, 1710)_ = 1.90, *p* = 0.012). Taken together, our results suggest that tamoxifen-induced caMEK5 expression in aNPCs may not increase the number of adult-born hippocampal neurons but does increase the dendritic complexity of adult-born neurons in both ApoE3-KI:caMEK5 and ApoE4-KI:caMEK5 mice.

## 3. Discussion

There is increasing research interest in understanding the contribution of genetic and environmental risk factors in AD pathogenesis and cognitive decline [4,29,30]. The links between Cd and AD in humans are limited to positive associations of markers of Cd exposure and mortality in AD patients in NHANES study cohorts [31,32]. We have previously reported direct toxicological evidence of GxE effect of ApoE4 and environmentally relevant concentrations of Cd [13] and lead [33] in ApoE4-KI mice. Additionally, these prior studies reported the GxE effect of ApoE4 and heavy metals on adult neurogenesis impairment, suggesting that adult neurogenesis may be an underlying mechanism for the GxE effect on cognitive function. However, our published observation of the GxE effect on cognitive function and adult neurogenesis impairments was correlative. To test our hypothesis that the GxE effect of ApoE4 and Cd on hippocampal adult neurogenesis contributes to the impairment of hippocampus-dependent learning and memory, we aimed to determine if the GxE impairment of learning and memory can be reversed by a conditional genetic enhancement of adult neurogenesis.

Here, we designed a functional rescue experiment to specifically and genetically stimulate adult neurogenesis once Cd-induced memory impairment was established in a GxE model utilizing ApoE3-KI:caMEK5 and ApoE4-KI:caMEK5 mice. This transgenic mouse model enabled us to selectively induce Cre-mediated transgenic recombination and caMEK5 expression through treatment with tamoxifen, thereby stimulating adult neurogenesis. We focused on male animals because male ApoE4-KI mice were more sensitive than females to the effects of Cd exposure [13]. Furthermore, our extensive effort of breeding the triple transgenic experimental animals with 198 breeding cages generated enough male but not female mice for both strains to use for both behavioral and cellular studies (see Section 4). Animals were treated with an environmentally relevant concentration of Cd through drinking water (0.6 mg/L CdCl_2_) throughout the experiment.

We have previously utilized the NOL test for longitudinal studies of heavy metal exposures on hippocampus-dependent learning and memory [13,21,33]. We used the NOL test in this study to probe hippocampus-dependent spatial memory performance through Cd exposure and tamoxifen treatment. ApoE4-KI:caMEK5 mice exhibited Cd-induced deficits in the NOL test earlier than ApoE3-KI:caMEK5 (onset at experimental week 16 vs. 28), suggesting that there is an effect of genotype between Cd-treated ApoE3-KI:caMEK5 and ApoE4-KI:caMEK5 mice on the onset of hippocampus-dependent spatial memory impairment. This finding is consistent with our previous report of male ApoE4-KI susceptibility to Cd-induced cognitive function impairment [13]. Although we lack a Cd-treatment control, our findings still support the presence of a GxE interaction between ApoE4 and Cd.

Following confirmation of NOL test impairment, we treated animals with tamoxifen or vehicle (control) to induce the expression of caMEK5 in aNPCs. The conditional and selective induction of caMEK5 was able to rescue Cd-induced NOL test impairment in ApoE4-KI:caMEK5 mice, as well as in ApoE3-KI:caMEK5 (genotype control). These findings suggest that caMEK5 expression in aNPCs is sufficient to rescue behavioral deficits; thus, stimulation of adult hippocampal neurogenesis through caMEK5 expression is directly linked to the genotype differences of Cd-induced hippocampus-dependent spatial memory. The rescue effect of tamoxifen treatment on NOL test impairment in ApoE3-KI:caMEK5 mice further suggests that adult neurogenesis is directly linked to Cd-induced impairments of hippocampus-dependent spatial memory even in our genotype control mice, which is consistent with our previous findings in caMEK5 mice [21]. Tamoxifen treatment may have some suppressive effects on neural progenitor cell proliferation and differentiation [26,34] and cognition [21,34], though some reports are inconsistent, possibly due to differences in animal age, treatment method, and time point of observation. Thus, our data likely presents an underestimate of the rescue effect of caMEK5 expression.

We further analyzed histological data from the cellular cohort of mice with a Cd-exposure and tamoxifen treatment timeline parallel to the behavioral cohort to investigate the effects of caMEK5 expression on adult neurogenesis. We previously reported that aNPC survival and the number of adult-born neurons are impaired in Cd-treated animals in caMEK5 mice [21] and ApoE4-KI mice [13]. We have also reported that caMEK5 expression restores aNPC survival as well as the number of adult-born neurons [21]. While we did not observe any effects of caMEK5 expression on aNPC survival, the number of adult-born neurons, or the number of adult-born immature neurons in the present study, showed a significant increase in the total dendritic length and dendritic crossings in both ApoE3-KI:caMEK5 and ApoE4-KI:caMEK5 mice treated with tamoxifen. This is consistent with our previous findings that caMEK5 expression enhances the dendritic complexity of adult-born neurons [25]. Dendritic complexity is used here as a proxy for adult-born neuron maturation, which is critical for neuron integration into existing hippocampal networks [20,21,33]. Unlike the two aforementioned studies in which we reported restoration of aNPC survival and the number of adult-born neurons, we continued Cd exposure of 0.6 mg/L CdCl_2_ in drinking water through the end of experiments in this study. Therefore, Cd exposure may have continued to impair the survival of aNPCs and adult-born neurons, and caMEK5 expression was not sufficient to increase the number of surviving aNPCs or adult-born neurons. Our behavioral data and cellular data together suggest that the observed rescue effects are directly linked to the recovery of dendritic complexity of adult-born neurons in both ApoE3-KI:caMEK5 and ApoE4-KI:caMEK5 mice. Taken together with our previous report on GxE effect of ApoE4 and Cd on hippocampus-dependent memory [13], our findings suggest a causal role of adult neurogenesis in the GxE effect of ApoE4 and Cd on hippocampus-dependent memory.

Tamoxifen treatment had no effect on locomotion or anxiety behavior in the open field test in ApoE4-KI:caMEK5 mice. On the other hand, tamoxifen treatment increased moving time and moving distance, and increased center entries and margin distance, indicating an increase in locomotion and a decrease in anxiety behavior in ApoE3-KI:caMEK5 mice. However, these differences in ApoE3-KI:caMEK5 may confound the NOL test rescue effect in tamoxifen-treated ApoE3-KI:caMEK5 mice because the increased activity may correlate with improved exploration.

We observed no differences in the blood or brain Cd levels except for an increase in brain Cd levels in tamoxifen-treated ApoE4-KI:caMEK5 mice. These data suggest that the functional rescue effects after tamoxifen administration are not due to tamoxifen causing less Cd accumulation in the blood or brain. We report in the present study that 47 and 40 weeks of exposure to 0.6 mg/L CdCl_2_ through drinking water resulted in 0.2–0.4 µg/L blood Cd levels for most ApoE3-KI:caMEK5 and ApoE4-KI:caMEK5 mice. These concentrations are within the range found in the general US population (men: 0.206–0.255 µg/L; women: 0.263–0.304 µg/L, 2011–2018 geometric mean range; [28]. Furthermore, we report the brain Cd levels to be approximately 1.5–2.0 pg/mg. While there are no large-scale reports of brain Cd levels in a general population to our knowledge, Cd has been detected in human brain tissue (ICP-MS: 4–30 pg/mg) in studies of autopsy tissue from general populations in countries in Central Europe, Scandinavia, and Asia [35,36,37]. We previously reported 0.7–1.3 pg/mg in ApoE3-KI and ApoE4-KI mice exposed to 0.6 mg/L CdCl_2_ through drinking water for 14 weeks [13] and approximately 2.0 ng/g in mice exposed to 0.6 mg/L CdCl_2_ through drinking water for 38 weeks followed by 29.5–32 weeks cessation of exposure, while control (0 mg/L CdCl_2_) animals had below 0.5 ng/g Cd in the brain [21]. In both previous reports as well as the present study, Cd-exposed animals exhibited impairments in hippocampus-dependent memory through an exposure paradigm representative of exposure levels in the general US population, thus providing biological plausibility of Cd neurotoxicity. Taken together with existing epidemiology reports of negative associations of Cd levels and cognitive function [9,10], Cd may affect cognitive functions at levels prevalent in the general US population.

While we report an exciting finding of an underlying mechanism of a GxE interaction effect, we caution against overinterpretation of our report as our study is limited to male mice and Cd-treated animals only due to the logistical limitations to obtain enough animal numbers for a more complete study including both sexes and Cd treatment control (Figure 1A). Despite these limitations, we report convincing evidence utilizing a gain-of-function mouse model that demonstrates the rescue of hippocampus-dependent memory and adult hippocampal neurogenesis following Cd impairment through the induced expression of caMEK5 amid continued exposure to Cd. Our continued exposure throughout the experiment may better reflect lifelong, low-level exposures in the general population, and our study sheds light on the possibility that even in life-long exposures to Cd, there may be therapeutic benefits from stimulation of adult neurogenesis for individuals with genetic risk factors like *APOE4*.

## 4. Materials and Methods

### 4.1. Animals

We utilized humanized ApoE3- and ApoE4-knockin (ApoE3-KI and ApoE4-KI) mice as a non-amyloid dependent genetic risk model of Alzheimer’s disease. These mice were provided by Dr. Nobuyo Maeda at the University of North Carolina, Chapel Hill [12] and maintained as homozygous lines in our animal facility.

We bred ApoE3-KI or ApoE4-KI mice with Nestin-CreER^TM^:caMEK5-eGFP^loxP/loxP^ (simplified as caMEK5) mice to obtain ApoE3-KI:caMEK5 and ApoE4-KI:caMEK5 mice. To obtain these triple transgenic animals for experimentation, we crossed Nestin-CreER^TM^:caMEK5-eGFP^loxP/loxP^:ApoE4-KI male mice with caMEK5-eGFP^loxP/loxP^:ApoE4-KI female mice to achieve a genotypic ratio of 1:1 (Figure 11A). The same breeding scheme was used to obtain ApoE3-KI:caMEK5. For both breeding procedures, male animals were less than 6 months old, and females were less than one year old. The ApoE3-KI:caMEK5 and ApoE4-KI:caMEK5 transgenic mouse models allow us to stimulate adult neurogenesis upon tamoxifen treatment through the expression of caMEK5 with the ApoE3-KI or ApoE4-KI background (Figure 11B). All animals in each experimental cohort were within 2 weeks of age. Following genotyping, male animals were randomly housed into groups of 4–5 animals per cage for the behavior cohort, or 2–4 animals per cage for the cellular cohort to balance age ranges across each treatment group, assigned a priori. Total animal numbers for the behavior cohort were n = 13–14 per treatment per genotype at the beginning of the experiment; n = 12–14 at the end of the experiment. Total animal numbers for the cellular cohort were n = 6–8 per treatment per genotype at the beginning of the experiment and n = 4–7 at the end of the experiment.

All mice were housed under standard conditions (12 h light/dark cycle) with food (Picolab Rodent Diet 20, Lab Diet, St. Louis, MO, USA) and autoclaved water (tap water purified by reverse osmosis, acidified with 2.4–2.8% HCl) provided ad libitum. All animal care and treatments were approved by the University of Washington Institutional Animal Care and Use Committee (protocol number 3041-04).

### 4.2. Cd Exposure

We have previously reported that Cd treatment with 0.6 mg/L CdCl_2_ in drinking water yields average blood Cd levels of 0.3–0.4 µg/L in C57BL/6 mice [21] and ApoE3-KI and ApoE4-KI mice [13], comparable to blood Cd levels in the general non-smoking US population. All mice were acclimated to water bottles for 2 weeks, then provided Cd (as 0.6 mg/L CdCl_2_) water at 8–10 weeks of age through the end of the experiment. Water bottles were replaced weekly. Water bottle weights were recorded prior to placement in the cage and following removal from the cage. The preparation, use, and disposal of hazardous reagents were conducted according to the guidelines from the Environmental Health and Safety Office at the University of Washington.

### 4.3. Study Design

The study design includes a behavior cohort (Figure 11C) and a cellular cohort (Figure 11D) that closely follows the timeline of the behavior cohort; for clarity, experimental timelines are described under “order of experiments”. For both behavioral and cellular experimental cohorts, all animals started their Cd exposure at 8–10 weeks of age and were exposed through the end of their experimental timelines at euthanization. Because novel object location (NOL) deficits following Cd exposure were expected to be observed at different time points for ApoE3-KI:caMEK5 animals and ApoE4-KI:caMEK5 animals, experimental timelines for tamoxifen administration differed by genotype accordingly.

The behavioral cohort animals underwent baseline behavior tests (open field test, NOL test) prior to Cd exposure. Following the start of Cd exposure, animals were probed for hippocampus-dependent spatial working memory using the novel object location test approximately every other week. Once NOL deficits were observed (as defined by the disappearance of significant differences between the percent of time spent in the familiar vs. novel locations), the deficits were confirmed by at least two subsequent tests one week apart. Once deficits were confirmed, animals were treated with tamoxifen to induce caMEK5 expression or vehicle. Following a recovery period from tamoxifen or vehicle treatment, animals were followed up with the open field and NOL tests. At the end of the experiment, brain and blood tissues were collected for Cd analysis.

No behavior experiments were conducted with the cellular cohorts. Experimental timelines for the cellular cohort were based on results in the behavior cohort, i.e., tamoxifen and vehicle treatment coincided with experimental weeks at which NOL deficits were confirmed in three consecutive weeks in the behavioral cohort, and tissue collection coincided with observation and confirmation of NOL rescue in the behavioral cohort.

### 4.4. Order of Experiments

For both ApoE3-KI:caMEK5 and ApoE4-KI:caMEK5 animals, tamoxifen treatment groups were assigned a priori. Baseline open field and NOL tests were conducted prior to Cd treatment (7–9 weeks of age). The start of the Cd exposure (8–10 weeks of age) was defined as experimental week 0. The timing of tamoxifen treatment differs for ApoE3-KI:caMEK5 and ApoE4-KI:caMEK5 animals due to their different onsets of memory deficits.

For ApoE4-KI:caMEK5 animals in the behavioral cohort, NOL tests were conducted at experimental weeks 2, 4, 6, 8, 10, 12, 14, 16, 17, 18, and 20, at which point NOL test deficits were confirmed. Animals underwent tamoxifen or vehicle treatment at experimental week 22. Following tamoxifen treatments, NOL tests were conducted at experimental weeks 32.5, 34, 38, and 39.5, and the open field test at week 41. Animals were euthanized at week 49. ApoE4-KI:caMEK5 animals in the cellular cohort were given tamoxifen or vehicle treatment at experimental week 22, then euthanized at week 40.

For ApoE3-KI:caMEK5 animals in the behavioral cohort, NOL tests were conducted at experimental weeks 2, 4, 6, 8, 10, 12, 14, 16, 18, 20, 22, 24, 25, 26, 28, 29, and 30, at which point NOL test deficits were confirmed. Animals underwent tamoxifen or vehicle treatment at week 32. Following tamoxifen treatment, NOL tests were conducted at experimental weeks 43, 45, and 47, and open field tests at week 48. Animals were euthanized at week 54. ApoE3-KI:caMEK5 animals in the cellular cohort were given tamoxifen or vehicle treatment at experimental week 32, then euthanized at week 47.

### 4.5. Open Field Test

The open field test was conducted before Cd exposure and after tamoxifen administration (experiment week 48 for ApoE3-KI:caMEK5; week 41 for ApoE4-KI:caMEK5) to assess the effects of Cd on locomotor activity and anxiety. Mice were placed in a 10 × 10 × 16 inch (width × depth × height) TruScan Photo Beam Tracking arena (Coulbourn Instruments, Whitehall, PA, USA) with clear Plexiglas sidewalls, and their movement was monitored with infrared beams with a spatial resolution of 0.3 inch. Each animal was allowed to freely explore the arena without prehabituation for 20 min, and the data were collected by TruScan 2.0 software (Coulbourn Instruments). The arena was cleaned with 5% acetic acid between experimental animals. The total number of moves, moving time, and moving distance were used to assess the effects of Cd and tamoxifen on locomotor activity. The number of center entries, time spent in the center and margin, and distance traveled in the center, and margin were used to assess the effects of Cd and tamoxifen on anxiety. The margin was defined as the area within 1.5 inches of the arena wall.

### 4.6. Novel Object Location Test

The novel object location (NOL) test was used to assess the effects of Cd on hippocampus-dependent spatial working memory. This assay was performed as previously described [13,21]. Briefly, each mouse was placed into an open field arena (Coulbourn Instruments) with two identical objects placed in two adjacent corners. During the training session, the mouse was allowed to freely explore the arena and objects, then returned to its home cage. One hour after training, the animal was returned to the arena with the same 2 objects, with one object in the original location and the other moved to a novel location. For each week of NOL test, object locations were randomized to exclude preference of specific locations and object pairs were randomly selected from 3 distinctly different shapes, all within 1.5 and 2 inches across (custom machined brass cone, spice jars filled with sand, and small liquor bottles filled with metal beads). Each training and test session lasted 5 min and was recorded by video cameras for later quantification. The time each animal spent actively investigating each object was manually scored and analyzed by an experimenter blinded to the animal’s treatment. The data inclusion criteria for NOL assays were a minimum of 0.5 s total exploration time. The data for treatment groups within each genotype were pooled prior to tamoxifen and vehicle treatment to increase the power of the NOL test analysis and lower the possibility of a type 1 error.

### 4.7. Tamoxifen and BrdU Administration

Following the confirmation of persistent spatial working memory deficits (3 consecutive deficits in the NOL test), animals were administered freshly prepared tamoxifen (200 mg/kg, dissolved in corn oil with 2% glacial acetic acid; Cat: B9285; MilliporeSigma, St. Louis, MO, USA) or vehicle (corn oil) by oral gavage once a day for 4 consecutive days in a cycle, for a total of 3 cycles with a 2-week interval between each tamoxifen treatment cycle, as previously described [25]. Tamoxifen induces the expression of caMEK5-eGFP in adult neural stem cells in caMEK5 mice through Cre-mediated recombination and expression of caMEK5-eGFP specifically in Nestin-expressing NPCs, allowing for the inducible and conditional activation of adult neurogenesis via caMEK5 activation of the endogenous ERK5 MAP kinase [25]. To identify adult-born cells, mice in the cellular cohort received two intraperitoneal injections of 100 mg/kg 5-Bromo-2′-deoxyuridine (BrdU; Cat: B9285; MilliporeSigma) per day for 3 consecutive days and euthanized 2.5 weeks later. Up to two animals in each tamoxifen-treated genotype group did not survive tamoxifen treatment, leading to the final animal numbers presented under the Animals section.

### 4.8. Immunohistochemistry

Mice in the cellular cohort were anesthetized with ketamine/xylazine and perfused intracardially with ice-cold solutions of 20 mL of PBS, followed by 20 mL of 4% paraformaldehyde (PFA) in PBS with heparin. Brains were collected and postfixed in 4% PFA/PBS overnight at 4 °C, followed by 30% (*w/v*) sucrose in PBS solution at 4 °C until brains sank. After sucrose embedding, brains were frozen at −80 °C until immunohistochemistry (IHC) processing. IHC was performed on 30 µm coronal brain sections using a free-floating antibody staining method as previously described [22,25]. All primary and secondary antibodies were diluted in blocking buffer (10% donkey serum, Cat: D9663, MilliporeSigma; 1% Bovine serum albumin, Cat: BAC61, Equitech-Bio, Kerrville, TX, USA). The primary and secondary antibodies and dilutions used in immunohistochemistry were rat monoclonal anti-BrdU (1:500; Cat: ab6326, abcam, Cambridge, UK), mouse monoclonal anti-NeuN (1:500; Cat: MAB 377, EMD Millipore, Burlington, MA, USA), goat polyclonal anti-DCX (1:200; Cat: SC-8066, Santa Cruz Biotechnology, Dallas, TX, USA), rabbit polyclonal anti-GFP (1:2000; Cat: A11122; Invitrogen, Carlsbad, CA, USA), Alexa Fluor Conjugated Secondary Antibodies (1:3000; A11077, A11001 or 1:500; A11055) and biotinylated goat anti-rabbit secondary antibody (1:250; BA1000, Vector Labs, Newark, CA, USA). After blocking and incubation with primary and secondary antibodies, tissues were incubated with 2.5 µg/mL Hoechst 33,342 (Cat: H3570, Invitrogen) for 30 min, washed 3× with PBS, then mounted onto slides using anti-fade Aqua Poly/Mount (Cat: 18606, Polysciences, Warrington, PA, USA) solution. For BrdU staining, the tissues underwent the following acid treatment prior to blocking: 10 min in 1N HCl at 4 °C, 30 min in 2N HCl at 37 °C, and 2 × 30 min in 0.1M borate buffer. For eGFP staining, a tyramide signal amplification kit (TSA; Cat: NEL701A001KT; Akoya Biosciences, Marlborough, MA, USA) was used as previously described [21,25].

### 4.9. Imaging and Quantification of Immunostained Cells

Immunostained cells from brain tissues were quantified as previously described [22,25]. One in every eight serial coronal brain sections containing the hippocampus (a total of 8 to 11 sections per brain) was immunostained for various markers and quantified by an experimenter blinded to genotype and treatment. Quantified numbers were multiplied by 8 to estimate the total number of marker-positive cells in the DG. The co-localization of positive markers was confirmed by overlapping fluorescent signals in a single cell using a Z-series stack. All images were captured with an Olympus Fluoview-1000 laser scanning confocal microscope (Olympus, Tokyo, Japan) with 10× or 20× lenses or with a Leica SP8X confocal microscope (Leica Microsystems, Buffalo Grove, IL, USA) with 10× (air), 20× (air), or 63× (oil) lenses. Optical Z-sections were processed and adjusted for color, brightness, and contrast uniformly using Fiji/ImageJ2 (version 2.9.0, NIH, Bethesda, MA, USA).

For dendritic morphology studies, 30 um sections stained only for DCX were used [38,39]. Cell tracing and Sholl analyses were performed blind to genotype and treatment using SNT (previously Simple Neurite Tracer; version 4.1.2., NIH) in the Neuroanatomy plugin as previously described [25,40]. DCX^+^ fluorescence signals were traced semi-automatically using SNT and were confirmed by the presence of continuous fluorescence signals in sequential X-Y, Y-Z, and X-Z cross-sections. 39–52 individual neurons were randomly chosen and analyzed from each genotype and treatment from 3–4 animals per group.

### 4.10. Blood and Brain Sample Collection and Analysis

Blood and brain samples were collected at the end of the experiment from the behavioral cohort for Cd measurement. Mice were anesthetized with ketamine/xylazine and checked for limb reflexes prior to cardiac puncture for blood collection (>0.3 mL/animal). Following a cervical dislocation, brain samples were dissected out, separated by hemisphere using a glass slide cover, snap frozen and stored at −80 °C until further analysis. Blood samples were stored at −20 °C until further analysis. Whole blood and right brain hemispheres were analyzed for brain Cd levels by the Environmental Health Laboratory at the University of Washington using inductively coupled plasma mass spectrometry. The experimenter who measured the blood and brain Cd was blinded to the treatment and genotype of animals. The Agilent 7900 (Agilent Technologies, Santa Clara, CA, USA) has a detection limit of approximately 0.3 ng/g per sample.

### 4.11. Statistical Analysis

All statistical analyses and data visualizations were conducted in R (version 4.1.3). For blood and cortex Cd concentrations, open field test, NOL tests, and total dendritic length, Welch’s two sample *t*-test (α = 0.05) were used for within-genotype comparisons in each treatment group (tamoxifen vs. vehicle). Wilcoxon rank sum exact tests were conducted for comparisons of BrdU^+^, NeuN^+^, and DCX^+^ cells within each genotype. Across-genotype comparisons were not performed due to timeline differences. For all figures presenting individual data points, a horizontal jitter is applied such that overlapping data points are more visible. For all figures presenting summaries of groups, all data are expressed as mean ± SEM.

Mixed-effects linear regression using restricted maximum likelihood estimation (α = 0.05) was used for longitudinal analysis of mouse body weight, water consumption, and Sholl analysis (adapted from [41]) with random effects from mouse ID. Type III ANOVA tables were constructed with Satterthwaite’s degree of freedom. *lme4*, *lmerTest*, and *emmeans* R packages were used for linear regression, ANOVA table construction, and pairwise comparisons (Welch’s two sample *t*-test) at specific time points or radii with Tukey’s HSD corrections.

## 5. Conclusions

Our present study found that specific and conditional stimulation of adult neurogenesis rescued Cd-induced impairments in hippocampus-dependent short-term spatial memory in a GxE model of ApoE4 and Cd, demonstrating a direct link between memory impairment and adult neurogenesis in this GxE model. Our findings provide strong evidence supporting the presence of a GxE effect and a potential underlying mechanism involving adult neurogenesis at Cd exposure levels relevant to the general US population. These findings support the hypothesis that environmental factors have considerable impacts on cognitive function and dementia risk [4,29], and show the potential benefit of interventions such as behavioral changes and policies targeting environmental factors on the cognitive health of the general population and on social burdens of AD and dementia. Furthermore, our findings highlight the potential of therapeutic interventions in mitigating age-related cognitive decline and dementia risk, particularly for individuals with elevated genetic risk, including easily accessible interventions such as exercise [14,42], diet [43], and lowering stress [44], which can stimulate adult neurogenesis.

## Figures and Tables

**Figure 1 ijms-24-09118-f001:**
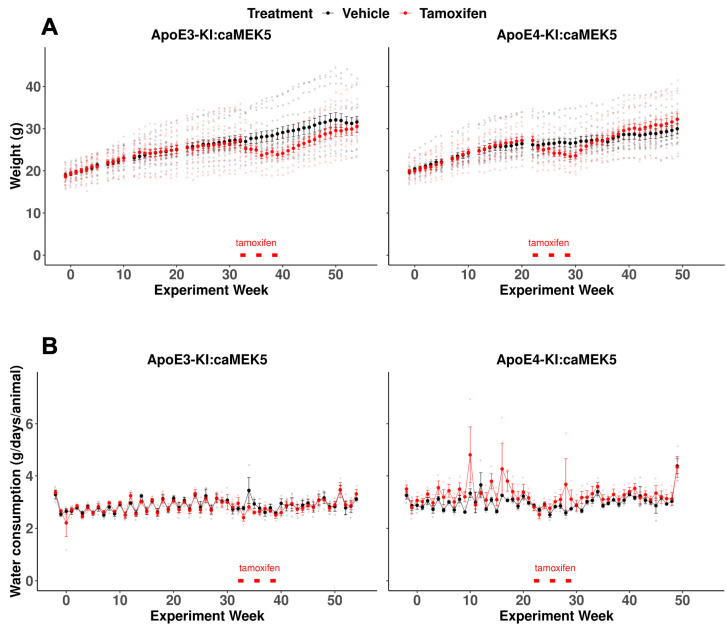
Animal weights and water consumption of the behavioral cohort for ApoE3:caMEK5 and ApoE4:caMEK5 mice. (**A**) Temporary body weight loss was observed in tamoxifen-treated ApoE3-KI:caMEK5 and ApoE4-KI:caMEK5 mice. (**B**) Tamoxifen treatment did not affect water consumption in ApoE3-KI:caMEK5 or ApoE4-KI:caMEK5 mice. Bars (red) indicate the time of tamoxifen treatment. Data are presented as mean ± SEM.

**Figure 2 ijms-24-09118-f002:**
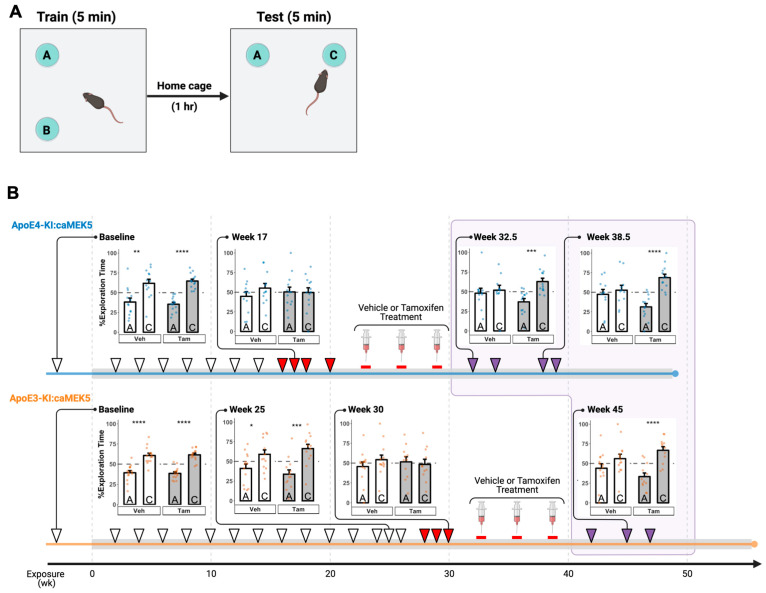
Effects of Cd exposure on hippocampus-dependent short-term spatial memory in the novel object location (NOL) test. (**A**) Schematic of NOL test. During training, two identical objects were placed in two adjacent locations A and B. The duration of time each mouse spent exploring identical objects in the old location A and novel location C was quantified in the test session. (**B**) ApoE4-KI and ApoE3-KI mice were unable to distinguish between the old and new locations starting at 16 and 28 weeks of Cd exposure, respectively. However, this impairment was reversed after tamoxifen, but not vehicle (veh) control treatment. Cd treatment lasted throughout the experiment (gray box); time points after vehicle/tamoxifen treatment are boxed in light purple. Each NOL test time point is indicated with an arrowhead. The observed NOL impairments are indicated by red arrowheads while restored memory tests are indicated by purple arrowheads. Representative data for % exploration time were shown for select time points (lines linked to the arrowheads). Data are presented as mean ± SEM. Welch’s two sample *t*-test: * *p* ≤ 0.05, ** *p* ≤ 0.01, *** *p* ≤ 0.001, **** *p* ≤ 0.0001.

**Figure 3 ijms-24-09118-f003:**
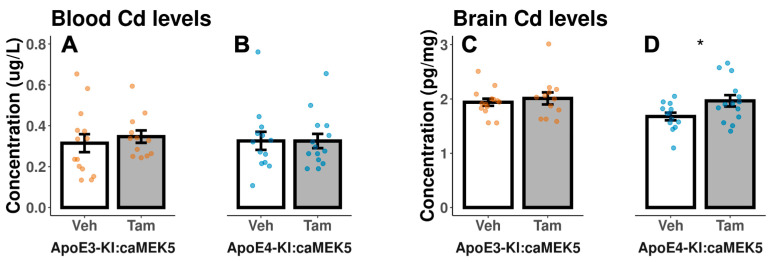
Cd concentrations in mouse blood and brain hemisphere at the end of the experiment in the behavioral cohort after 54 weeks (ApoE3-KI:caMEK5) or 49 weeks of exposure (ApoE4-KI:caMEK5). There were no significant differences between vehicle- and tamoxifen-treated groups in (**A**) blood Cd of ApoE3-KI, (**B**) blood Cd of ApoE4-KI, or (**C**) brain Cd of ApoE3-KI. (**D**) Tamoxifen-treated ApoE4-KI mice had statistically significant higher brain Cd levels compared to vehicle-treated ApoE4-KI mice. Data are presented as mean ± SEM. Welch’s two sample *t*-test: * *p* ≤ 0.05.

**Figure 4 ijms-24-09118-f004:**
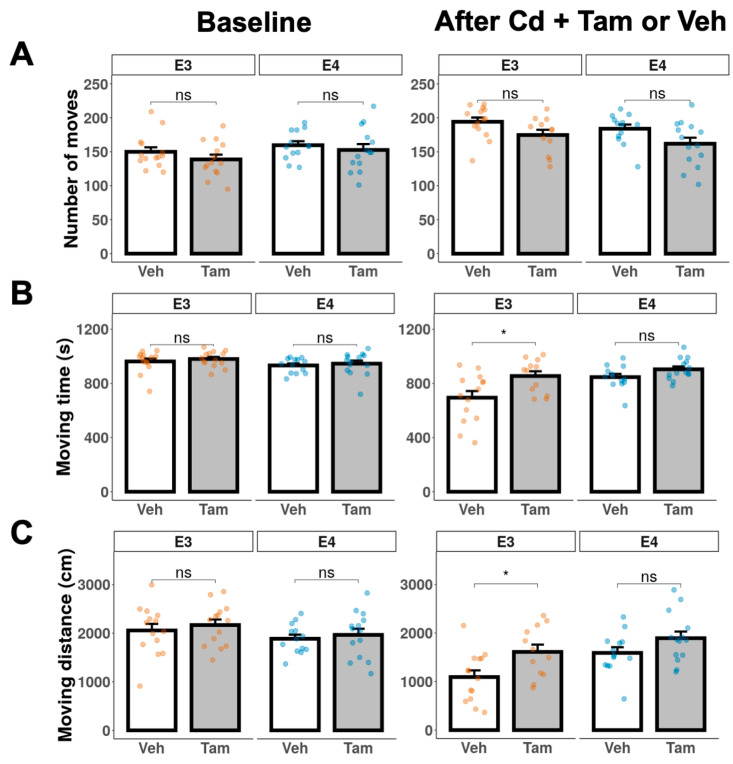
Locomotor activity of ApoE3-KI and ApoE4-KI mice (n = 12–15/group) was measured in the open field test at baseline and after Cd + Veh or Tam treatment. (**A**) number of moves. (**B**) total moving time. (**C**) total moving distance. Data are presented as mean ± SEM. Welch’s two sample *t*-test: * *p* ≤ 0.05.

**Figure 5 ijms-24-09118-f005:**
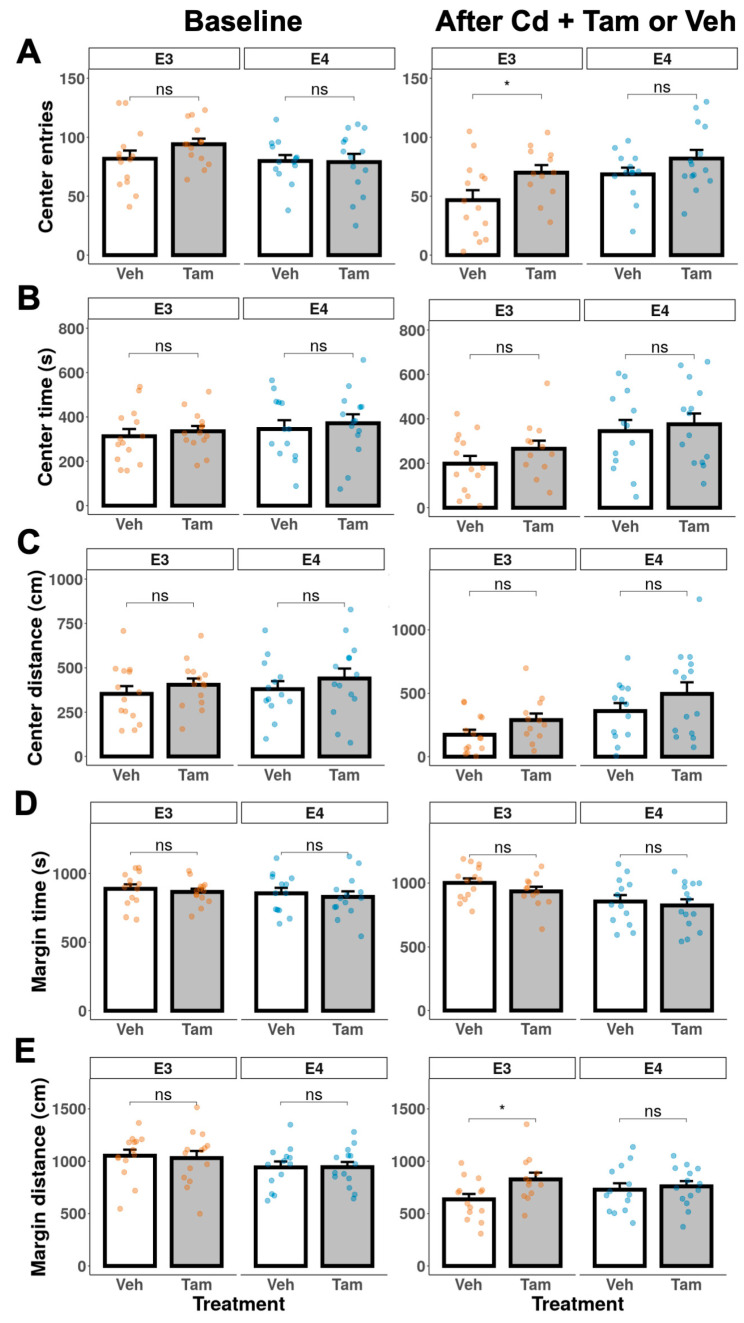
Anxiety behavior of ApoE3-KI and ApoE4-KI mice (n = 12–15/group) was measured in the open field test at baseline and after Cd + Veh or Tam treatment. (**A**) number of entries to the center, (**B**) center time, (**C**) center distance, (**D**) margin time, or (**E**) margin distance. Data are presented as mean ± SEM. Welch’s two-sample *t*-test: * *p* ≤ 0.05.

**Figure 6 ijms-24-09118-f006:**
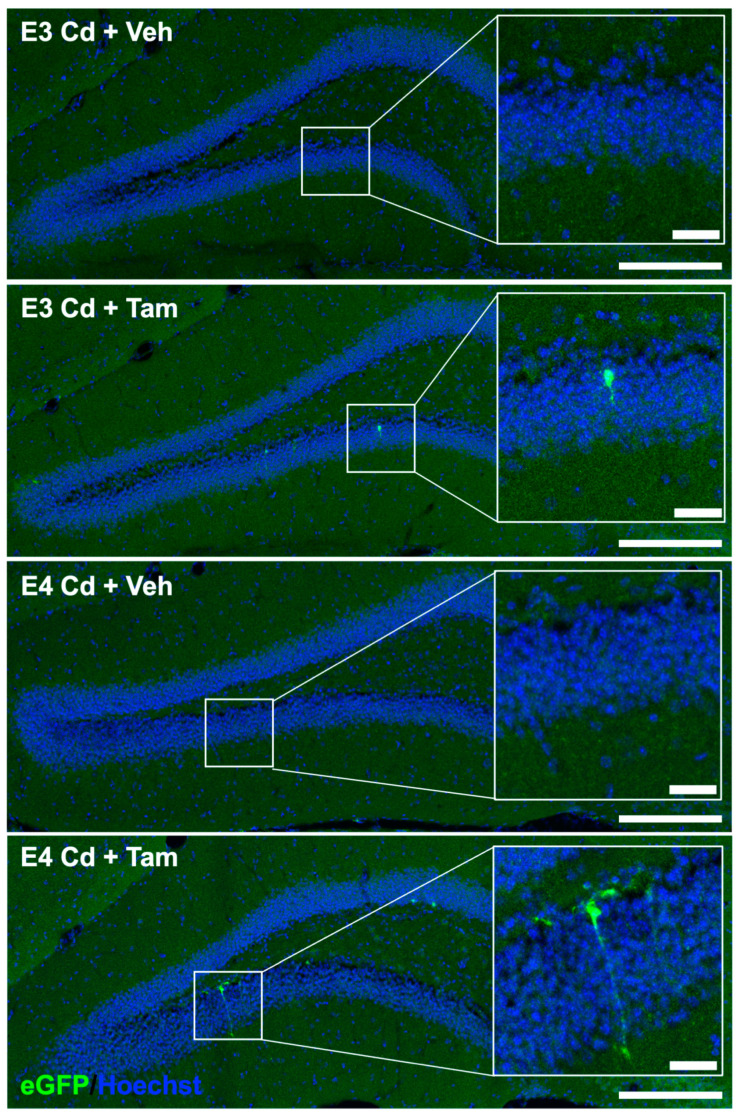
Representative images of TSA signal amplified caMEK5-eGFP signal. Full DG field images were taken on a 20× objective and stitched in LAS X; scale bar: 200 µm. Inlay: Zoomed-in images from ROI; scale bar: 30 µm.

**Figure 7 ijms-24-09118-f007:**
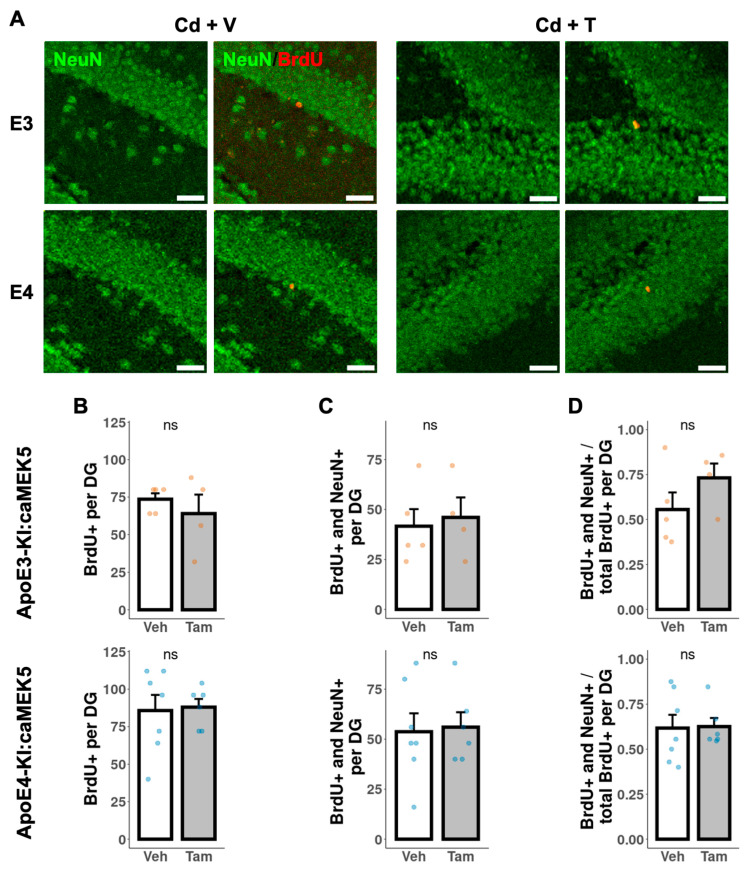
Quantification of adult-born neurons that differentiated into mature neurons in the DG of the hippocampus. (**A**) Representative images of BrdU^+^ (red) and NeuN^+^ (green) staining. Scale bar, 30 µm. Quantification of (**B**) total number of BrdU^+^ cells, (**C**) total number of BrdU^+^ and NeuN^+^ cells, and (**D**) ratio of BrdU^+^ and NeuN^+^ cells to BrdU^+^ cells for tamoxifen- or vehicle-treated ApoE3-KI:caMEK5 (n = 4–5) and ApoE4-KI:caMEK5 (n = 6–7) mice. Data are presented as mean ± SEM.

**Figure 8 ijms-24-09118-f008:**
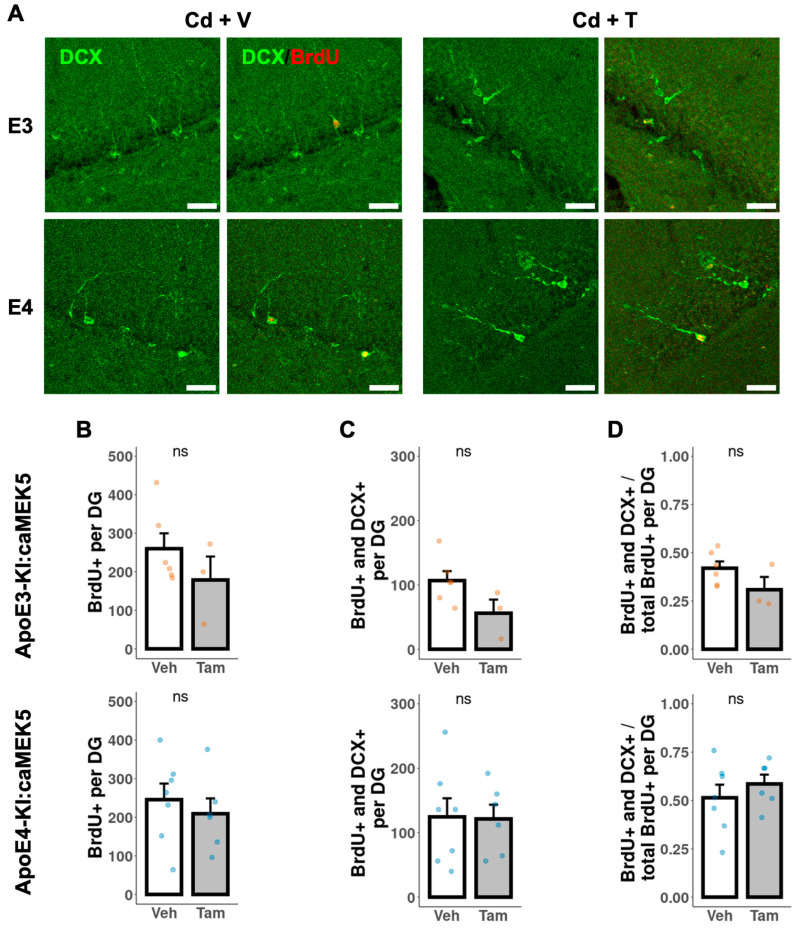
Quantification of adult-born immature neurons in the DG of the hippocampus. (**A**) Representative images of BrdU^+^ (red) and DCX^+^ (green) staining. Scale bar, 30 µm. Quantification of (**B**) total number of BrdU^+^ cells, (**C**) total number of BrdU^+^ and DCX^+^ cells, and (**D**) ratio of BrdU^+^ and DCX^+^ cells to BrdU^+^ cells for tamoxifen- or vehicle-treated ApoE3-KI:caMEK5 (n = 4–5) and ApoE4-KI:caMEK5 (n = 6–7) mice. Data are presented as mean ± SEM.

**Figure 9 ijms-24-09118-f009:**
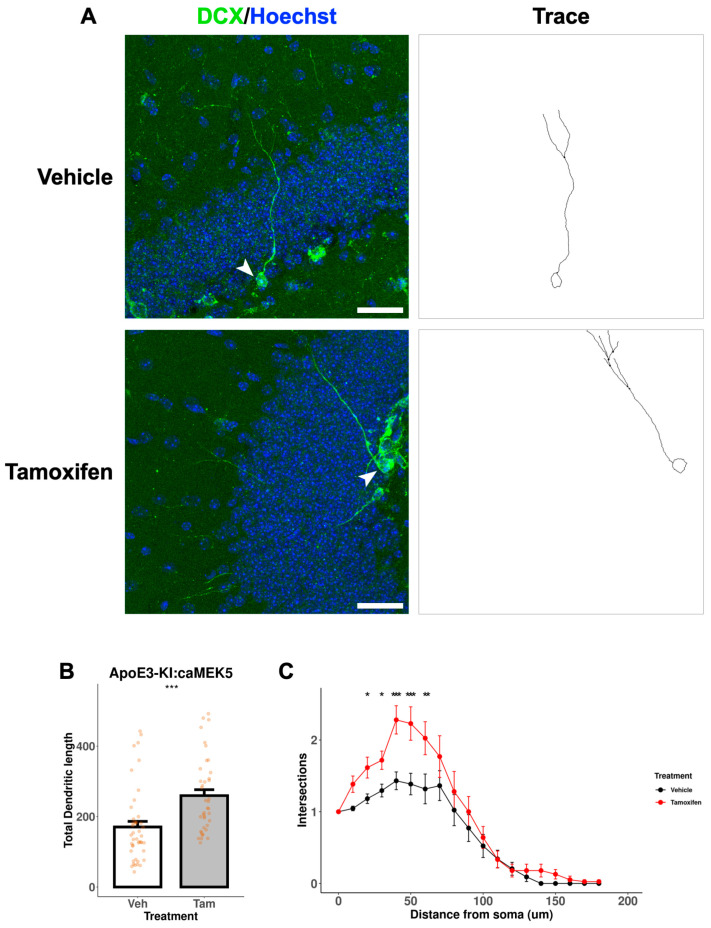
Dendritic branching of DCX^+^ cells in the DG of ApoE3-KI:caMEK5 mice. (**A**) Representative images (arrowhead) of cell tracing of immature neurons with Hoechst (blue) and DCX (green) staining in the DG. Scale bar, 30 µm. (**B**) Tamoxifen-treated animals have significantly increased total dendritic length compared to vehicle-treated animals. Welch’s two sample *t*-test: *** *p* ≤ 0.001. (**C**) Tamoxifen-treated animals have significantly increased numbers of intersections compared to vehicle-treated animals. Pairwise comparisons at each experimental week were based on estimates from mixed-effects linear regression, with Tukey HSD correction: * *p* ≤ 0.05, ** *p* ≤ 0.01, *** *p* ≤ 0.001. n = 3–4 animals per group, 39–44 cells per group.

**Figure 10 ijms-24-09118-f010:**
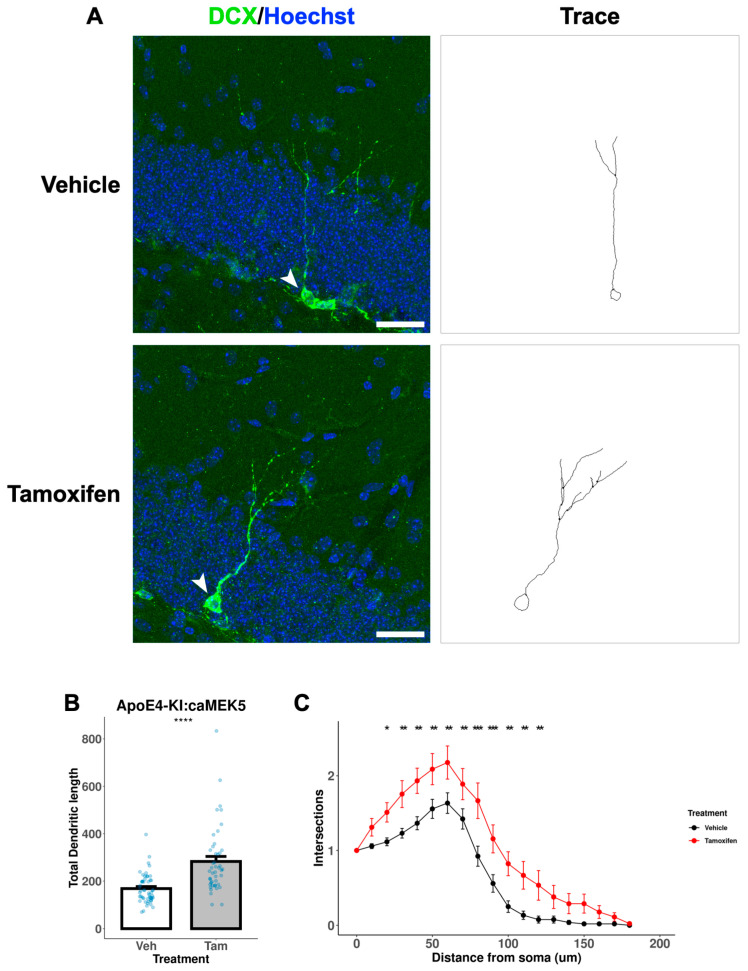
Dendritic branching of DCX^+^ cells in the DG of ApoE4-KI:caMEK5 mice. (**A**) Representative images (arrowhead) of cell tracing of immature neurons with Hoechst (blue) and DCX (green) staining in the DG. Scale bar, 30 µm. (**B**) Tamoxifen-treated animals have significantly increased total dendritic length compared to vehicle-treated animals. Welch’s two sample t-test: **** *p* ≤ 0.0001. (**C**) Tamoxifen-treated animals have significantly increased numbers of intersections compared to vehicle-treated animals. Pairwise comparisons at each experimental week were based on estimates from mixed-effects linear regression, with Tukey HSD correction: * *p* ≤ 0.05, ** *p* ≤ 0.01, *** *p* ≤ 0.001. N = 4 animals per group, 45–52 cells per group.

**Figure 11 ijms-24-09118-f011:**
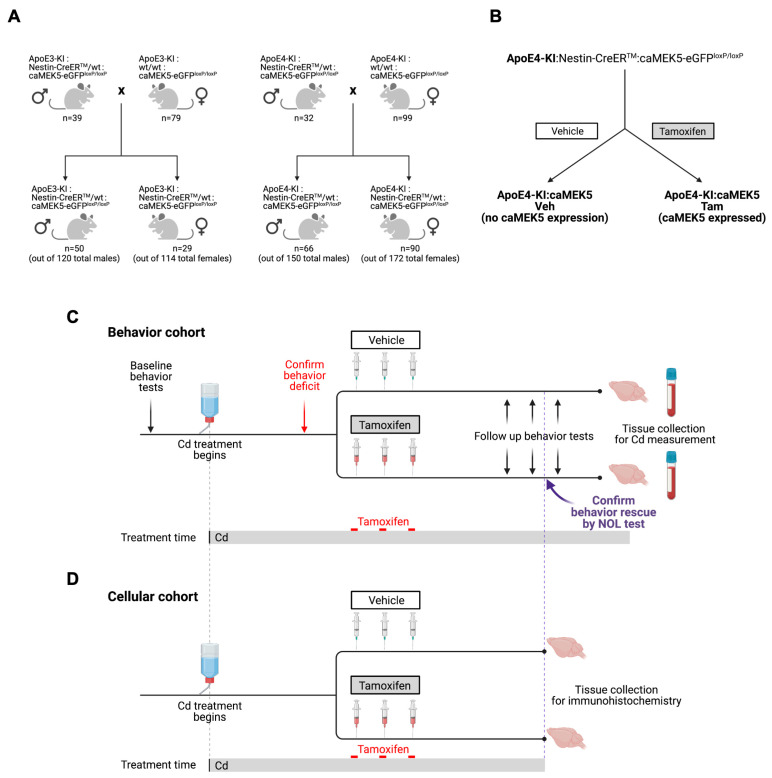
Study design for the caMEK5 rescue experiment. (**A**) Breeding scheme. (**B**) The schematic diagram for generation of transgenic mice through tamoxifen treatment. Tamoxifen-treated animals express constitutively active MEK5 (caMEK5) in adult neurogenic regions (under the Nestin-CreER^TM^ promoter) with ApoE4-KI background (ApoE4-KI:caMEK5); control animals have the same genetic background but do not express caMEK5 (ApoE4-KI:control). The same strategy was used to generate ApoE3-KI:caMEK5 and control mice. (**C**) Study design for the caMEK5 rescue experiment within each genotype in the behavior cohorts. Prior to Cd treatment, baseline behavior was assessed with the open field and novel object location (NOL) tests. All animals received Cd through drinking water (0.6 mg/L CdCl_2_) from 8–10 weeks of age until the time of euthanasia. Cognitive function was probed every other week with a NOL test until a deficit was observed, at which time, NOL was performed weekly. Once memory deficits were confirmed by the NOL test at three consecutive time points, animals were given tamoxifen (n = 12–15/group) treatment to induce caMEK5 expression. The control group received a vehicle that was used to dissolve tamoxifen and thus should not express caMEK5. Four weeks after the last dose of tamoxifen or vehicle administration, the NOL test was performed to assess cognitive function after tamoxifen or vehicle treatment. The open field test was performed after the functional rescue was confirmed in at least three consecutive NOL tests. At the end of the experiment, blood and brain tissues were collected for Cd analysis. (**D**) Study design for the caMEK5 rescue experiment within each genotype in the cellular cohorts. Tissue collection in the cellular cohort was based on the last date of the three consecutive time points of NOL rescue confirmation in the behavior cohort.

## Data Availability

The data presented in this study are available in this article. Additional data are available from the corresponding author upon request.

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
