# Peer review of "Inducible and Conditional Activation of Adult Neurogenesis Rescues Cadmium-Induced Hippocampus-Dependent Memory Deficits in ApoE4-KI Mice"

_ijms, 2023, doi:10.3390/ijms24119118_

Round 1
Reviewer 1 Report
Journal: IJMS
Manuscript ID: ijms-2382715
Type: Article
Title: Inducible and conditional activation of adult neurogenesis rescues cadmium-induced hippocampus-dependent memory deficits in ApoE4-KI mice
Authors: Megumi T. Matsushita, Hao Wang, Glen M. Abel, and Zhengui Xia *
Comments to the Author(s)
In the manuscript titled " Inducible and conditional activation of adult neurogenesis rescues cadmium-induced hippocampus-dependent memory deficits in ApoE4-KI mice '' the Authors use a mouse knock-in model of a TAM inducible caMEK5 that stimulates adult neural stem/progenitor cells, to reverse the effect of Cd-induced memory deficits in two mouse strains that express the ApoE3 and ApoE4 alleles which are genetic risk factors for late-onset Alzheimer’s Disease. The caMEK5 transgene is driven from a nestin promoter which allows its conditional activation of hippocampal dentate gyrus neurogenesis.
For each of the ApoE3-Ki:caMEK5 and ApoE4-Ki:caMEK5 groups, the Authors use physiological mouse data (body weight and water consumption), behavioral analysis (new object location, locomotor activity and anxiety), blood and brain Cd levels, as well as imaging analysis of caMEK5-eGFP, BrdU+ , NeuN+, and DCX staining. to show the following: a) Cd levels at physiological concentrations in human smokers impair mouse performance in the NOL test without affecting overall locomotor activity; the effect is present in both ApoE4-Ki:caMEK5 and ApoE3-Ki:caMEK5 groups albeit the former is more susceptible than the latter, with a faster induction (20 weeks v 30 weeks); b) the effect is reversed by caMEK5 induction upon TAM administration; c) the effect is not caused by differences in brain Cd clearance before and after caMEK5 induction; d) caMEK5 expression stimulates adult hippocampal neurogenesis but does not statistically significantly increase the number of adult-born immature or mature-differentiated neurons in the dentate gyrus; and e) caMEK5 induction by TAM increases dendritic branching in both ApoE4-Ki:caMEK5 and ApoE3-Ki:caMEK5 groups.
Overall evaluation: the Authors should be commended for a well performed study. Experimental hypotheses are grounded in the literature and well formulated, controls are carefully considered and issues related to ApoE allele-dependent differences in time-course are well explained.The diagrams are very helpful and provide sufficient details. The Methods section is similarly well-written.
Strengths: this is a very competent study, well performed and analyzed, appropriately statistically powered and well described. The Authors are very meticulous in their discussion of the experimental variables, controls and conclusion, without overreaching in their discussion.
Weaknesses: none.
Suggestions: please place the appropriate figure references in the Results section.
Reviewer 2 Report
The manuscript by Matsushita et al. describes whether stimulation of adult hippocampal neurogenesis functionally rescues Cd-induced cognitive impairment in knockin (KI) mice, a mouse model of Alzheimer’s disease (AD). Transgenic mice used in the current study expressed the human ε4 (ApoE4-KI) or human ε3 alleles (ApoE3 KI) under the control of the endogenous mouse apolipoprotein E (ApoE) gene promoter. The APoE gene is known to be a genetic risk factor for late-onset AD. These transgenic mice were crossed with Nestin-CreERTM:caMEK5-eGFPloxP/loxP double transgenic mice to generate triple-transgenic mouse lines that enable conditionally and genetically stimulation of adult hippocampal neurogenesis. The data related to mouse body weight and water consumption, blood and cortex Cd concentrations, and behavior of animals in the novel object location test and open field test are presented elegantly in graphical form. However, the data concerning morphology of newborn neurons in the dentate gyrus are not properly analyzed and documented. In particular, I'm wondering if it is possible to apply Sholl analysis using a 30 µm thickness brain section. To study dendritic arbor complexity, thicker sections of brain tissue are usually used (between 80 and 120 µm or more). The second concern is the poor quality of images illustrating doublecortin (DCX) immunostaining. All images visualizing DCX+ immature neurons should be replaced with new, high-quality confocal images. Furthermore, in Fig. 8 A, DCX-immunopositive cells in ApoE3-KI:caMEK5 treated with tamoxifen appear to have lower dendritic arborization complexity than those of vehicle-treated ApoE3-KI:caMEK5 mice. I also noticed that DCX-labeled cells, shown in Fig. 9, have green soma and red dendrites. Please, clarify this.
Reviewer 3 Report
Authors investigated the gene (ApoE3 and ApoE4)-environment interaction on hippocampal functions in cadmium-intoxicated mice based on behavioral and morphological analysis. The results are interesting, but there are some concerns to consider for publication.
Authors analyzed the morphological data with BrdU/NeuN and DCX in the dentate gyrus. Authors treated BrdU 2.5 weeks before euthanizatio and they observed BrdU positive cells in the subgranular zone of dentate gyrus. However, it should be moved into granule cell layer.
How many DCX-positive neuroblasts were analyzed for this study? The number of DCX-positive neuroblasts may be few and it may be various stages of neuroblasts.
In this study, authors described that they used goat polyclonal anti-DCX (1:200; Cat: SC-8066, Santa Cruz Biotech), but this antibody is sold out about 7 years ago.
Authors should check the reference found in line 101, 123, 128, 132, 145, and 152 (Error! Reference source not found).
Authors should check the some spell errors.
Round 2
Reviewer 2 Report
Point 1: The authors responded to criticisms concerning Sholl analysis using a 30-μm-thick brain section by referring to their previous work. Are there any papers by other authors that provide such an analysis on 30-μm-thick brain sections?
Point 2: I agree that “the acid treatment process involved in anti-BrdU staining can negatively impact dendritic staining”. However, the authors should clarify why the acid treatment in BrdU staining only selectively damages the neuronal processes in tamoxifen-treated ApoE3-KI:caMEK5 and not vehicle-treated ApoE3-KI:caMEK5 mice (Figure 8).
Point 3: In Figure 9, the DCX+ cell in the DG of the vehicle-treated mouse has more dendrites than shown (Fig. 9 A Trace).
Reviewer 3 Report
The manuscript has been improved and I have no further comment.
Authors should check the minor spelled errors.
Author Response
We thank you for your helpful and constructive comments.